# Interdisciplinary Multimodal Pain Rehabilitation in Patients with Chronic Musculoskeletal Pain in Primary Care—A Cohort Study from the Swedish Quality Registry for Pain Rehabilitation (SQRP)

**DOI:** 10.3390/ijerph20065051

**Published:** 2023-03-13

**Authors:** Lukasz Mateusz Falkhamn, Gunilla Stenberg, Paul Enthoven, Britt-Marie Stålnacke

**Affiliations:** 1Department of Community Medicine and Rehabilitation, Rehabilitation Medicine, Umeå University, SE-901 87 Umeå, Sweden; 2Department of Community Medicine and Rehabilitation, Physiotherapy, Umeå University, SE-901 87 Umeå, Sweden; 3Department of Medical and Health Sciences, Linköping University, SE-581 83 Linköping, Sweden

**Keywords:** interdisciplinary rehabilitation, chronic pain, register studies, disability

## Abstract

Chronic pain is a major public health issue. Mounting evidence suggests that interdisciplinary multimodal pain rehabilitation programs (IMMRPs) performed in specialist pain care are an effective treatment for patients with chronic pain, but the effects of such treatment if performed in primary care settings have been less studied. The aims of this pragmatic study were to (1) describe characteristics of patients participating in IMMRPs in primary care; (2) examine whether IMMRPs in primary care improve pain, disability, quality of life, and sick leave 1-year post discharge in patients with chronic pain; and (3) investigate if outcomes differ between women and men. Data from 744 (645 women and 99 men, age range 18–65 years) patients with non-malignant chronic pain included in the Swedish Quality Registry for Pain Rehabilitation Primary Care were used to describe patient characteristics and changes in health and sick leave. At 1-year follow-up, the patients had improved significantly (*p* < 0.01) in all health outcome measures and had reduced sick leave except in men, where no significant change was shown in physical activity level. This study indicates that MMRPs in primary care improved pain and physical and emotional health and reduced sick leave, which was maintained at the 1-year follow-up.

## 1. Introduction

Chronic pain (>3 months) [1] is a major public health issue, and about 25–35% of adults in Europe suffer from chronic pain [2]. The prevalence of chronic pain increases with age, and women are disproportionally affected [3,4]. Chronic pain often affects several areas in life such as reduced participation in activities at home and in society [5], and at least half of the people with chronic pain report that it interferes with their work [2].

Chronic pain has an impact on and interacts with various social, physical, emotional, and psychological factors [6] such as insomnia, anxiety, depression, low level of physical activity, fatigue, psychomotor inhibition, and weight gain, which lead to poor quality of life [5,7]. More women than men report high pain intensity, and pain has a greater negative impact on life both at home and in society for women compared with men [8].

Although chronic pain has a significant impact on both individuals and society, Breivik et al. (2006) found that up to 50% of persons with chronic pain had not received adequate treatment for their pain [5].

Since chronic pain is a complex condition, a biopsychosocial (BPS) framework is used in interdisciplinary multimodal pain rehabilitation programs (IMMRPs), which take into account biological, psychological, and social/contextual factors [4,5]. IMMRPs require coordinated treatment modalities from several professions. The team work is integrated using an interdisciplinary approach. Patients participate actively in collaboration with the team in terms of the planning, goal setting, and fulfilment of the rehabilitation period [6]. IMMRPs combine psychosocial interaction, pedagogical efforts, and physical activity in a program. Since chronic pain has a number of negative consequences, one of the purposes of IMMRP is for the patient to learn how to use adaptive coping strategies and develop their capacity to do so. The primary goal is for the patient to become more involved in everyday activities such as return to work and thereby reduce sick leave, while the secondary goal is to improve physical ailments such as pain [9].

Research mostly based on studies in specialist care has shown that IMMR is an effective and evidence-based intervention for persons with chronic pain, reducing sick leave and improving return to work [6,10,11,12,13]. Results are heterogeneous, with some patients showing improvement in a variety of areas, while others do not improve at all [10,11,12,13,14]. 

IMMR is a well-established method in specialist care in Sweden for patients with severe complex chronic pain. Since most patients with chronic pain are seen in primary care and because up to 40% of primary care visits are made up of patients with chronic pain [9], the Swedish government introduced in 2009 a national care guarantee for primary care. The intention was to stimulate the development of IMMRP at the primary care level. The rehabilitation guarantee ensured that healthcare providers received special financial compensation for patients with non-specific chronic pain who completed IMMRP. The goal was primarily to reduce sick leave and secondarily to reduce pain [9]. National guidelines with selection criteria were published to support assessment of patients with chronic pain to enhance IMMR to appropriate level (specialist care vs. primary care [9]. The guarantee ended in 2020, but IMMRPs without this financial compensation are still available and implemented in primary care in some county councils in Sweden. Patients participating in these IMMRPs are registered in the Swedish Quality Register for Pain Rehabilitation in primary care (SQRP-pc) [15]. However, little research has been done on IMMR in primary care. It is important to continue to develop IMMRPs and increase knowledge about patients with chronic pain in primary care. 

Chronic pain is more frequently reported among women than men [16], but there are also more women than men being referred to IMMRP [17]. In previous studies from both specialist care [17,18,19] and primary care [14,20,21], fewer men than women were included. Several possible explanations for this can be identified; e.g., when women and men experience the same symptoms, they express it in different ways [22]. In qualitative studies, professionals have expressed that men are rarely given a chronic pain diagnosis [21], and doctors hesitate to give a diagnosis such as chronic pain to men because these diagnoses are considered unmanly. Furthermore, men are to a greater extent than women recommended unimodal rehabilitation such as physiotherapy and also radiological examinations [14]. It is important to study the benefits of IMMRPs in relation to different patient characteristics, for example, gender [23], in order to be able to design programs that would benefit all patients and thereby bring about equal care. Research has given rise to inconsistent results regarding different outcomes for women compared with men after IMMRPs. Some studies show no differences [24]. Other studies show that IMMRP is more beneficial for women than for men [19,20,25], while still others show that men benefit more in some variables [26]. 

Thus, there is a gap in knowledge regarding the long-term results of IMMR in primary care on a national level. The aims of this study were to (a) describe characteristics of patients with chronic pain participating in IMMRPs in primary care; (b) to examine if patients participating in IMMRPs improve regarding pain aspects, emotional and physical functioning, coping, health-related quality of life, and sick leave at 1-year follow-up; and (c) investigate whether outcomes for IMMRPs differ between women and men. 

## 2. Materials and Methods

### 2.1. Study Design

This was a registry study with a 1-year follow-up based on data obtained from the SQRP-pc [15]. SQRP-pc data are stored with the approval of the National Swedish Data Inspection Agency (permission number 1580-97). The study followed the ethical principles of the Declaration of Helsinki and Swedish law regarding the use of personal data [27,28] and was approved by the Ethical Review Board in Linköping, Sweden (Dnr: 2017/483-31). 

### 2.2. Participants and Setting

This study included SQRP-pc data from patients ≥18 years old with chronic (≥3 months), nonmalignant pain who were referred to primary healthcare centers from 2016 to the spring of 2021 and answered the baseline questionnaire. Inclusion criteria for IMMRP were (i) age between 18–65 years; (ii) disabling, nonmalignant chronic pain (on sick leave or experiencing major interference in daily life due to chronic pain); (iii) no further medical investigations needed; and (iv) written consent to participate in and attend IMMRP. Exclusion criteria for IMMRP were severe psychiatric comorbidity, abuse of alcohol and/or drugs, diseases that did not allow physical exercise, and specific pain conditions associated with red flags. These criteria follow national guidelines for IMMRP in primary care, which also require that pharmacological treatment and non-pharmacological unimodal rehabilitation, e.g., physiotherapy, have been tested without satisfactory effect [29].

The IMMRPs were based on a bio-psycho-social approach with interdisciplinary teamwork and included goalsetting together with the patient and interventions such as physical exercise, relaxation, training in coping strategies based on cognitive behavioral therapy (CBT), and education in pain management. The IMMRPs lasted for a limited time (6–10 weeks, 1.5–3 h/week) with several meetings a week and were carried out in groups, individually, or as a mixture of both group and individual sessions. The composition of the team varied depending on the health center. A team could consist of a general practitioner, a sociologist, a psychologist, a physiotherapist, an occupational therapist, a nurse, and a dietitian in different combinations and to different degrees of involvement. Both the content of the IMMRP and the professional competencies were according to the national guidelines for IMMRP in primary care [18].

### 2.3. Measures (and Procedure)

The variables and instruments used in this study are mandatory for the primary care rehabilitation centers registering their data with the SQRP-pc. They cover important outcome domains for the evaluation of chronic pain, as recommended by the Initiative on Methods, Measurement, and Pain Assessment in Clinical Trials (IMMPACT) [30] and the validation and application of patient-relevant core outcome set to assess the effectiveness of multimodal pain therapy (VAPIAN) [31]. Most patient-reported outcome measures (PROMS) were collected through questionnaires at baseline, after the IMMRP, and at the 1-year follow-up post IMMRP. Leisure-time physical activity was collected at baseline and at the 1-year follow-up.

#### 2.3.1. Background Characteristics

The following background characteristics were extracted from the SQRP-pv: age (years); gender (man or woman); education level (compulsory school, upper secondary/vocational school, university/college); and country of birth (Sweden, Nordic country outside of Sweden, European but non-Nordic country, and non-European country). Data on household (living with parents and siblings, cohabitant or partner, and/or children under 18 y); employment status (working, jobseeker, or student); and income support (yes/no) were also extracted. Expectations of IMMR were reported on a 4-point scale (fully recovered, some improvement, not recovered but relief obtained, or no expectation of either recovery or relief). Number of visits to a doctor during the last year was reported at 4 levels (none, 1–2 times, 2–3 times, or 4 times or more).

#### 2.3.2. Pain Aspects

Current pain intensity and pain intensity over the last 7 days were marked on an 11-point numeric pain rating scale (NPRS), with 0 representing “no pain” and 10 “worst pain imaginable”. NPRS 0–5 was defined as mild pain, 6–7 as moderate pain, and 8–10 as severe pain [32]. Pain variation was reported as recurrent or persistent pain. 

The number of pain sites was registered using 36 pre-defined anatomical areas. The patient reported the number of sites with pain on the left side of the body (*n* = 18) and on the right side of the body (*n* = 18) for a total of 36 locations. These pain sites were (1) head/face, (2) neck, (3) shoulder, (4) upper arm, (5) elbow, (6) forearm, (7) hand, (8) anterior aspect of chest, (9) lateral aspect of chest, (10) belly, (11) sexual organs, (12) upper back, (13) lower back, (14) hip/gluteal area, (15) thigh, (16) knee, (17) shank, and (18) foot. The total score ranges between 1–36.

Duration of pain was reported as years with pain (number of days since pain started and the number of days with persistent/chronic pain).

#### 2.3.3. Emotional and Physical Functioning

The hospital anxiety and depression scale (HADS) contains 14 items: 7 items to measure anxiety (HADS-A) and 7 items to measure depression (HADS-D) [33,34,35]. Each item is rated from 0 to 3 indicating how bothersome the problem was during the last week. The total score for the subscales ranges between 0 and 21, with a higher score indicating a worse condition. The obtained scores can be divided into 3 categories where a score of 0–7 indicates no anxiety/depression, a score of 8–10 a mild disorder, and a score of 11 or higher is the cut-off for a possible clinically significant disorder [35].

The functional rating index (FRI) measures activity and participation in relation to the International Classification of Functioning Disability and Health (ICF) [36]. The FRI consists of 10 items about, for example, walking, lifting, sleep, and activities of daily living. The answers are graded on a 5-point scale. The sum of these items is converted to a percentage, where 0% indicates full self-rated function, and 100% means that the patient does not perceive any function at all [36]. Many patients with chronic musculoskeletal pain have chronic low-back pain or chronic neck pain and often pain in both locations. The FRI is recommended for the assessment of disability in people with multi-area spinal pain [37]. Furthermore, the FRI has shown good responsiveness for patients with chronic low-back pain [38] and chronic neck pain [39].

The Godin–Shephard leisure-time physical activity questionnaire (GSLTPAQ) captures how many times a week a respondent performs different leisure activities and at what level: strenuous, moderately strenuous, or light exercise. The participant is also asked how many times on average they engage in physical activity that leads to an increase in pulse rate. The points are added together to make a total score. Strenuous activity is multiplied by 9, while light activity is multiplied by 3 [40,41].

#### 2.3.4. Coping

The pain catastrophizing scale (PCS) consists of 13 items describing different thoughts and feelings when experiencing pain [42]. The PCS instructions ask participants to reflect on painful experiences on a 5-point scale from 0 (not at all) to 4 (all the time). The PCS total score (patient’s degree of pain-related catastrophizing) ranges from 0–52, in which 52 signifies maximal catastrophizing. A score of <24 on the PCS was reported as low, and a score ≥ 24 on the PCS was reported as high pain catastrophizing [43]. The PCS also contains scores for rumination, magnification, and helplessness, but in this study, only the total score was used.

#### 2.3.5. Health-Related Quality of Life

The EQ-5D European quality of life instrument measures HRQoL [17]. The European quality of life instrument (EQ-5D) [44,45,46] consists of two parts: the EQ-5D (3-L) and the EQ-VAS. The EQ-5D (3-L) contains five dimensions: mobility, self-care, usual activities, pain/discomfort, and anxiety/depression. Each dimension has three levels: no problems, some problems, and extreme problems. The answers on the five dimensions are converted into a single EQ-5D index ranging from −0.594 to 1, where 1 indicates optimal health. The EQ-VAS records the respondent’s self-rated health on a vertical visual analogue scale that ranges from 0 (“worst possible health state”) to 100 (“best possible health state”).

#### 2.3.6. Sick Leave

Sick leave was reported as full time (100%) or part time (25–75%). 

### 2.4. Statistics

Descriptive data were presented as mean with standard deviation (SD), median and interquartile range (IQR), or numbers and percentages. Comparisons between women and men were analyzed using unpaired *t*-test, Mann–Whitney U Test, chi-square test, and chi-square test of trend (linear-by-linear association). Change over time between baseline, after MMR, and at 1-year follow-up was analyzed using paired *t*-test, Wilcoxon signed-rank test, and McNemar’s test. Effect sizes with 95% confidence intervals (95% CI) on comparison between women and men and change over time were calculated using the website Psychometrica [35], and the values were evaluated against Cohen’s criteria, which state that the absolute ES of 0.0–0.2 has no significance, 0.2–0.49 has small, 0.5–0.79 has medium, and ≥0.8 has strong significance [47]. Results with *p*-value < 0.05 were statistically significant. The statistics were processed in IBM Statistical Package for the Social Sciences (version 24.0 SPSS Inc., Chicago, IL, USA).

## 3. Results

The SQRP-pc included patient data from 49 primary healthcare centers in Sweden. Baseline patient characteristics obtained from the SQRP-pc are reported in Table 1. Most of the 4357 patients were women (*n* = 3542, 81.3%; men: *n* = 815, 18.7%). The number of years of pain duration varied, and the mean was 7.4 ± 8.3 years. Of the 4357 patients registered in the SQRP-pc, 2809 (64.5%) answered the questionnaire after completion of the IMMRP. Of the patients who filled in the questionnaire after IMMRP, 26.5% also filled in the questionnaires after one year. In total, 744 (17.1%) patients filled in all three questionnaires at baseline, after MMR, and at 1-year follow-up (Table 1).

Regarding background factors, there were some differences between those who completed the 1-year follow-up SQPR-pc compared with those who did not. Among completers, there was a higher proportion of women (18.1%) than men (12.2%, *p* = <0.001), they were older (mean (SD) 44.7 (10.1) vs. 43.5 (12.2) years, respectively, *p* = <0.010), a higher proportion was born in Sweden (86% vs. 79.4%, respectively, *p* = <0.001), more had a high education (14% vs. 11%, respectively, *p* = 0.039), more were working (81.5% vs. 70%, respectively, *p* = <0.001), and fewer had income support (3.5% vs. 6.3%, respectively, *p* = 0.009). 

Regarding pain variables, there were also some differences between those who completed the 1-year follow-up SQPR-pc compared with those who did not. Among completers, there was a higher proportion with persistent pain (77.4% vs. 71.3%, respectively, *p* = <0.001), a lower proportion of completers rated their pain intensity as severe (32.0% vs. 32.0%, *p* = <0.001), and completers reported more pain sites (15.4 (7.8) vs. 13.9 (8.2), respectively, *p* = <0.001).

Regarding emotional and physical functioning, completers rated lower on PCS (25.2 (10.8) vs. 26.3 (11.6), respectively, *p* = 0.015) compared with those who did not complete the follow-up. Completers also had a lower proportion of patients with clinical anxiety (26.1% vs. 28.3%, respectively, *p* = <0.001) and clinical depression (22.0% vs. 23.6%, respectively, *p* = <0.001).

Health-related quality of life differed as well, where completers rated higher on EQ-5D-VAS (43.8 (20.4) vs. 42.2 (18.3), respectively, *p* = 0.048) and had a higher EQ-5D-index (0.37 (0.32) vs. 0.33 (0.33), respectively, *p* = 0.017) compared with those who did not complete the follow-up.

In patients who answered the questionnaires at baseline, after IMMRP, and at 1-year follow-up, most outcome measures improved between baseline and after IMMRP (effect size small to medium) and to a somewhat lesser extent between baseline and 1-year follow-up (see Table 2). Godin 7 days and Godin activity level did not change significantly. For the whole group, there were no significant changes between the PROMS right after IMMRP and 1-year follow-up except that the FRI improved (mean (SD) 1.7 (14.5), *p* = 0.002), and HADS depression deteriorated (−0.5 (3.5), *p* < 0.001).

There were no statistically significant differences between women and men in most of the PROMS at the three reporting time points, such as pain variables, pain catastrophizing, physical activity during last seven days, and EQ-5D index. Women reported lower FRI compared with men at all three time points (*p* = 0.020, 0.026, and 0.013, respectively; see Appendix Table A1). Women reported higher HADS anxiety compared with men at baseline (*p* = 0.018). Women reported lower HADS depression compared with men at 1-year follow-up (*p* = 0.010, not in Table 2a), and HADS depression worsened from right after IMMRP to 1-year follow-up more in men than in women (*p* = 0.024). Women had a lower Godin activity level compared with men at baseline (*p* = 0.047). EQ-VAS was higher in women compared with men after IMMRP and at 1-year follow-up (*p* = 0.042 and 0.003, respectively), and EQ-VAS also improved more in women compared with men between baseline and 1-year follow-up (*p* = 0.035).

Sick leave decreased significantly between baseline and 1-year follow-up both for all patients and for women and men separately (*p* < 0.001; see Table 3). There were no statistically significant differences in sick leave between women and men at baseline (*p* = 0.570) and at 1-year follow-up (*p* = 0.154).

## 4. Discussion

In this study, PROMS in participants with chronic pain were studied before, after, and at 1-year follow-up after IMMRP in primary care. The results indicate improvements in physical and emotional function, pain intensity, physical activity, and quality of life. In general, stronger effect sizes were shown for women than for men. In addition, the degree of sick leave was reduced at 1-year follow-up for both women and men.

In this study, an improvement was seen in the outcome variables pain, emotional function, and physical function for both sexes. This improvement was maintained over time. This is in contrast to a recently published study [13,23] from specialist care, where an improvement was seen in both sexes immediately after IMMRP, but the change only remained for women at the 1-year follow-up. However, participants in the study from specialist care were patients only from the northern part of Sweden, while our study in primary care included patients from the whole country. There were also significant differences between women and men in baseline data in our study that were not seen in the study by Spinord et al. [23], e.g., in age and employment.

In terms of patient characteristics, in this study, the majority of the patients were born in Sweden, had attended at least upper secondary school, and had persistent pain. These results are consistent with a Swedish study by Pietilä-Holmner et al. [20] based on SQRP-pv data from two county councils, but in contrast to that study, this study showed poorer mental health and poorer quality of life. This may indicate that patients at a national level have a lower level of mental health and therefore rate worse on these outcome measures. When compared with some studies from specialist care [13,48], the present study showed similar proportions of country of birth, level of education, main occupation, and type of pain characteristics. However, in contrast, they showed better quality of life and a shorter period of chronic pain. This variation in both primary care and specialist care shows that it might be difficult for a physician to choose the right level of care despite the fact that there are national guidelines for this [21].

When the national care guarantee was introduced for primary care, its purpose was to stimulate the development of IMMRP at the primary care level and to direct more patients to IMMRP through financial incentives. The goal was primarily to reduce sick leave and secondarily to reduce pain [9]. In this study, a statistically significant reduction was seen in the degree of sick leave for both women and men. This is in line with other studies from both primary care [20,49] and specialist care [19,50], but there are also studies that did not show any change [13,51]. The discrepancy that exists in the literature may be because the studies were performed in different time periods using data from different geographical areas. Over the years, the social insurance system for sickness benefit has changed [52]. For studies in Sweden, where IMMRPs’ effect on sick leave has been evaluated, several factors have been identified that affect sick leave and return to work. In a recent study from specialist care [53], it was seen that high sickness absence, low physical activity, long pain duration, and low health-related quality of life before IMMRP increased the risk of full-time sick leave at the 1-year follow-up. The study concluded that it is important that patients receive an earlier intervention to reduce the probability of sick leave even after IMMRP.

Most systematic reviews based on data from specialist care have concluded that IMMRP is an effective method for treating chronic pain [6,10,11]. The findings in the current study indicate improvements in terms of pain, emotional and physical function, physical activity, and reduced sick leave after participating in IMMRP. This differs from the results found by Pietilä-Holmner et al. from 2020 [20], who saw a gender difference. In their study, it was found that women improved significantly in all outcome variables, while men only had an improvement in function level, which showed a medium ES. That study also showed that ES values were generally better for women than for men. The differences between the studies can be explained by the fact that this study had a larger population. Unlike their study, there are other studies in primary care that showed positive effects of IMMRP for men [54,55]. Compared with studies in specialist care, this study shows similar positive effects in terms of pain intensity and emotional function both in the long and short term for both genders [13] but also similar ES values (small and medium) at 1-year follow-up [56,57]. As previously discussed, the fact that more women than men participate in IMMRP can influence outcomes. In this study, the results for men showed generally lower ES compared with women. Based on this, one can speculate whether IMMRP is better adapted for women and that they therefore benefit more from it than men do. Previous studies have shown that gendered stereotyped norms among professionals can influence rehabilitation, for example, notions that MMR is better suited for women than men [21,58].

In this study, there are some limitations. There was no control group and thus no alternative treatment. This may have affected the interpretation of the results, as it cannot be ruled out that the results of this study are due to natural causes. However, most patients in this study had experienced pain for several years, and chronic pain is unlikely to resolve itself, which has also been proven [59]. There was a large drop-out of patients, which makes it difficult to generalize the results. However, this is in line with other studies that also used data from SQRP [20,57]. IMMRP in primary care is a relatively new intervention, which means the routines for reporting the questionnaires are not established in the same way as they are in special care. In this study, it was found that the questionnaire respondents were more often women, were older, more of them had been born in Sweden, were better educated, and were more often working. They reported more frequent persistent pain but fewer pain sites and less severe pain, they had less anxiety and depression, rated their health higher, had higher health-related quality of life, and less pain-catastrophizing thoughts than those who dropped out.

We do not know if the patients who did not complete the questionnaires were assessed by professionals as not being suitable to participate in the program, if they were sent to specialist care, or if they dropped out of their own free will. The difference between these groups gives rise to reflections about what norms (both professional and patient norms) prevail regarding which patient characteristics are suitable for IMMRP programs. Are the programs designed to suit all patients? A qualitative study in primary care expressed opinions about which patients are suitable for participation in a group program but also stated that it is impossible to know in advance who will benefit from MMRP [21].

There may be reason to reflect on which patients we address with IMMRP and which patients can be attracted and benefit from the scheme. Lehti et al. [58] found that patients with higher educational levels were perceived by professionals as being easier to interact with, and thoughts about gender norms influenced the rehabilitation options. Another reason for the dropouts may partly be that those who were not helped tended to be less likely to respond.

Although the competencies of the professionals were according to the national guidelines, the composition of the team varied depending on the health center. We can therefore not rule out that this may have affected the treatment and could be considered a confounding variable.

Since all questionnaires are self-assessments, including the question regarding sick leave, misunderstandings can easily arise, as some patients do not know if they have been entitled to sick leave by the Social Insurance Office. In addition, it must be considered that the requirements for receiving sick leave have changed over the years. For these reasons, the positive results of reducing the degree of sick leave should be interpreted with some caution. A major strength of the study was that this is the first study to investigate IMMRP in primary care on a national level based on the SQRP-pc. Previous studies [14,20,49] have studied smaller areas in primary care. With a larger population, it is easier to generalize data for primary care. Even though statistically significant positive results were shown, the clinical relevance of these findings are of most importance. Chronic pain is a complexity of biological, psychological, and social factors. Different parts of the biopsychosocial model of pain affect the individuals’ experiences and consequences of pain. In order to investigate the outcomes in patients participating in IMMRP, we used measures that covered different aspects of chronic pain. The outcome variables are in agreement with the biopsychosocial model and based on a validated and well-used questionnaire, which has been recommended both by Initiative on Methods, Measurement, and Pain Assessment in Clinical Trials (IMMPACT) [60] as well as the validation and application of patient-relevant core set of outcome domains to assess interdisciplinary multimodal therapy (VAPIAN) [31].

## 5. Conclusions

This study indicates that patients with chronic pain may benefit from IMMRP in primary care settings in terms of pain intensity, physical and emotional function, physical activity, health-related quality of life, and reduction in sick leave both in the short and long term.

The large number of dropouts, i.e., the decrease in numbers of those who participated at the beginning versus those who participated at the end, highlights the importance of strengthening the routines for the follow-up of patients participating in IMMRP in primary care.

## Figures and Tables

**Table 1 ijerph-20-05051-t001:** Patient characteristics. Comparison between those who answered the patient-reported outcome measures (PROMS) at 1-year follow-up (included cases) or not and between women and men.

	All (*n* = 4357)					Women (*n* = 3542, 81.3%)					Men (*n* = 815, 18.7%)					Difference Women vs. Men in Complete Cases
	Complete Cases (Included) (*n* = 744, 17.1%)		Cases with Missing Data (Not Included) (*n* = 3613, 82.9%)			Complete Cases (Included) (*n* = 645, 18.2%)		Cases with Missing Data (Not Included) (*n* = 2897, 81.8%)			Complete Cases (Included) (*n* = 99, 12.1%)		Cases with Missing Data (Not Included) (*n* = 716, 87.9%)			
	*n*	(%)	*n*	(%)	*p*-Value	*n*	(%)	*n*	(%)	*p*-Value	*n*	(%)	*n*	(%)	*p*-Value	*p*-Value
Age, mean (SD)	44.8	(10.9)	43.6	(12.2)	**0.009**	44.5	(10.9)	43.6	(12.1)	0.095	47.0	(11.2)	43.5	(12.6)	**0.008**	**0.022**
Country of origin, *n* = 4337					**<0.001 ^1^**					**0.004 ^1^**					**0.043 ^1^**	0.372 ^1^
Sweden	638	(86.0)	2855	(79.4)		549	(85.4)	2295	(79.6)		89	(89.9)	560	(78.7)		
Nordic countries	13	(1.8)	67	(1.9)		13	(2.0)	52	(1.8)		0	(0.0)	15	(2.1)		
Europe	27	(3.6)	190	(5.3)		23	(3.6)	156	(5.4)		4	(4.0)	34	(4.8)		
Non-European	64	(8.6)	483	(13.4)		58	(9.0)	380	(13.2)		6	(6.1)	103	(14.5)		
Education, *n* = 4131					0.519 ^2^					0.242 ^2^					0.064 ^2^	**<0.001 ^2^**
University	219	(30.4)	1083	(31.8)		202	(32.3)	898	(32.9)		17	(17.9)	185	(27.1)		
Upper secondary	422	(58.6)	1850	(54.2)		362	(57.9)	1457	(53.4)		60	(63.2)	393	(57.6)		
Elementary school	79	(11.0)	478	(14.0)		61	(9.8)	374	(13.7)		18	(18.9)	104	(15.2)		
Relationships sharing a household					NA					NA					NA	
Parents and siblings	19	(2.6)	134	(3.7)		14	(2.9)	98	(4.6)		5	(5.1)	36	(5.0)		
Cohabitant or partner	551	(74.1)	2229	(61.7)		487	(75.5)	1802	(62.2)		64	(64.6)	427	(59.6)		
Children under 18	270	(36.3)	1267	(35.1)		239	(37.1)	1057	(36.5)		31	(31.3)	210	(29.3)		
Employment status																
Working	606	(81.5)	2528	(70.0)	**<0.001 ^1^**	534	(82.8)	2057	(71.0)	**<0.001 ^1^**	72	(72.7)	471	(65.8)	0.170 ^1^	**0.016 ^1^**
Jobseeker	114	(15.3)	598	(16.6)	0.407 ^1^	95	(14.7)	442	(15.3)	0.733 ^1^	19	(19.2)	156	(21.8)	0.555 ^1^	0.251 ^1^
Student	26	(3.5)	261	(7.2)	**<0.001 ^1^**	23	(3.6)	218	(7.5)	**<0.001 ^1^**	3	(3.0)	43	(6.0)	0.229 ^1^	0.787 ^1^
Recurrent or persistent pain					**<0.001 ^1^**					**0.002 ^1^**					0.184^1^	0.571 ^1^
Recurrent pain	165	(22.6)	992	(28.7)		145	(23.0)	811	(29.2)		20	(20.4)	181	(26.7)		
Persistent pain	564	(77.4)	2462	(71.3)		486	(77.0)	1965	(70.8)		78	(79.6)	497	(73.3)		
Years since debut of pain, mean (SD)	9.4	(9.9)	9.2	(9.5)	0.620	9.4	(9.9)	9.3	(9.5)	0.903	9.6	(9.8)	8.8	(9.5)	0.409	0.771
Income support					**0.009 ^1^**					**0.008 ^1^**					0.648 ^1^	0.514 ^1^
Yes	21	(3.5)	176	(6.3)		17	(3.3)	141	(6.4)		4	(4.8)	35	(6.0)		
No	572	(96.5)	2616	(93.7)		492	(96.7)	2069	(93.6)		80	(95.2)	547	(94.0)		
Expectations of IMMRP ^a^					0.887 ^2^					0.984 ^2^					0.593 ^2^	0.530 ^2^
Fully recovered	112	(15.3)	647	(18.6)		92	(14.5)	491	(17.6)		20	(20.8)	156	(22.8)		
Some improvement	379	(51.8)	1659	(47.7)		341	(53.6)	1375	(49.2)		38	(39.6)	284	(41.6)		
Not to be recovered but to obtain relief	212	(29.0)	966	(27.8)		181	(28.5)	788	(28.2)		31	(32.3)	178	(26.1)		
No expectation of either recovery or relief	29	(4.0)	208	(6.0)		22	(3.5)	143	(5.1)		7	(7.3)	65	(9.5)		
Visits to a doctor last year					0.540 ^2^					0.943 ^2^					0.095 ^2^	0.057 ^2^
0–1 time	130	(17.6)	658	(18.6)		119	(18.6)	528	(18.7)		11	(11.1)	130	(18.6)		
2–3 times	207	(28.0)	980	(27.8)		180	(28.1)	797	(28.2)		27	(27.3)	183	(26.1)		
4 or more	403	(54.5)	1893	(53.6)		342	(53.4)	1506	(53.2)		61	(61.6)	387	(55.3)		

Abbreviations: NA, not applicable. ^1^ Chi-square test. ^2^ Chi-square test of trend. *p*-values < 0.05 presented in bold.

**Table 2 ijerph-20-05051-t002:** **(a)** Change in patient-reported pain, pain catastrophizing, anxiety, and depression between baseline and after IMMRP (1–2), between after IMMRP and 1-year follow-up (2–3), and between baseline and 1-year follow-up (1–3) ^1^. Comparison between women and men. (**b**) Change in patient-reported function, physical activity, and health-related quality of life between baseline and after IMMRP (1–2), between after IMMRP and 1-year follow-up (2–3), and between baseline and 1-year follow-up (1–3) ^1^. Comparison between women and men.

(a)																							
	All							Women							Men								Difference Women vs. Men
	Baseline (1)		After IMMRP (2)			1-Year Follow-Up (3)		Baseline (1)		After IMMRP (2)			1-Year Follow-Up (3)		Baseline (1)		After IMMRP (2)			1-Year Follow-Up (3)			*p*-Value
Pain NRS last week, *n* = 727, 628, 99																							0.110 ^3^
Mild (≤5)	186 (25.6)		336 (46.2)			340 (46.8)		156 (24.8)		286 (45.5)			296 (47.1)		30 (30.3)		50 (50.5)			44 (44.4)			
Moderate (6–7)	323 (44.4)		277 (38.1)			236 (32.5)		277 (44.1)		240 (38.2)			198 (31.5)		46 (46.5)		37 (37.4)			38 (38.4)			
Severe (8–10)	218 (30.0)		114 (15.7)			151 (20.8)		195 (31.1)		102 (16.2)			134 (21.3)		23 (23.3)		12 (12.1)			17 (17.2)			
Change (1–2) *p*-value ^2^ Effect size (CI-95%) ^3^	**<0.001** 0.41 (0.25–0.45)							**<0.001** 0.40 (0.23–0.45)							**<0.001** 0.42 (0.11–0.67)								0.884
Change (2–3) *p*-value Effect size (CI-95%)			0.105 −0.06 (−0.16–0.04)							0.243 −0.04 (−0.15–0.08)							0.137−0.14 (−0.42–0.14)						0.406
Change (1–3) *p*-value Effect size (CI-95%)						**<0.001** 0.33 (0.19–0.40)							**<0.001** 0.35 (0.19–0.42)							**0.010** 0.26 (−0.02–0.54)			0.534
Pain NRS current, *n* = 719, 620, 99																							0.335 ^3^
Mild (≤5)	272 (38.1)		434 (60.4)			409 (56.9)		230 (37.1)		370 (59.7)			355 (57.3)		44 (44.4)		64 (64.6)			54 (54.5)			
Moderate (6–7)	281 (28.9)		208 (28.9)			197 (27.4)		248 (40.0)		185 (29.8)			170 (27.4)		33 (33.3)		23 (23.2)			27 (27.3)			
Severe (8–10)	164 (22.8)		77 (10.7)			113 (15.7)		142 (22.9)		65 (10.5)			95 (15.3)		22 (22.2)		12 (12.1)			18 (18.2)			
Change (1–2) *p*-value Effect size (CI-95%)	**<0.001** 0.49 (0.33–0.54)							**<0.001** 0.45 (0.25–0.48)							**<0.001** 0.43 (0.12–0.68)								0.874
Change (2–3) *p*-value Effect size (CI-95%)			**0.002** −0.11 (−0.21–0.01)							**0.012** −0.53 (−0.62–0.40)							**0.041** −0.22 (−0.51–0.05)						0.087
Change (1–3) *p*-value Effect size (CI-95%)						**<0.001** 0.30 (0.16–0.37)							**<0.001** 0.32 (0.16–0.39)							0.076 0.18 (−0.1–0.46)			0.205
PCS, *n* = 716, 618, 98																							0.353 ^3^
Low (<24)	308 (43.0)		473 (66.1)			474 (66.2)		262 (42.4)		411 (66.5)			413 (66.8)		46 (46.9)		62 (63.3)			61 (62.2)			
High (≥24)	408 (57.0)		243 (33.9)			242 (33.8)		356 (57.6)		207 (33.5)			205 (33.2)		52 (53.1)		36 (36.7)			37 (37.8)			
Change (1–2) *p*-value Effect size (CI-95%)	**<0.001** 0.47 (0.24–0.45)							**<0.001** 0.52 (0.38–0.61)							**0.002** 0.32 (0.02–0.58)								0.078
Change (2–3) *p*-value Effect size (CI-95%)			0.937 0.0 (−0.10–0.10)							0.864 0.0 (−0.11–0.11)							0.835 −0.02 (−0.30–0.26)						0.641
Change (1–3) *p*-value Effect size (CI-95%)						**<0.001** 0.44 (0.20–0.41)							**<0.001** 0.49 (0.32–0.55)							**0.007** 0.28 (−0.03–0.53)			0.167
HADS anxiety, *n* = 730, 631, 99																							**0.018 ^3^**
None (≤7)	275 (37.7)		398 (54.5)			406 (55.6)		225 (35.7)		343 (54.4)			350 (55.5)		50 (50.5)		55 (55.6)			56 (56.6)			
Mild (8–10)	171 (23.4)		162 (22.2)			150 (20.5)		154 (24.4)		138 (21.9)			129 (20.4)		17 (17.2)		24 (24.2)			21 (21.2)			
Severe (11–21)	284 (38.9)		170 (23.3)			174 (23.8)		252 (39.9)		150 (23.8)			152 (24.1)		32 (32.3)		20 (20.2)			22 (22.2)			
Change (1–2) *p*-value Effect size (CI-95%)	**<0.001** 0.42 (0.33–0.54)							**<0.001** 0.44 (0.34–0.56)							**0.006** 0.31 (0.14–0.70)								0.189
Change (2–3) *p*-value Effect size (CI-95%)			0.826 −0.01 (−0.12–0.09)							0.772 0.0 (−0.11–0.11)							0.890 −0.01 (−0.29–0.26)						0.227
Change (1–3) *p*-value Effect size (CI-95%)						**<0.001** 0.40 (0.28–0.49)							**<0.001** 0.41 (0.29–0.51)							**0.042** 0.22 (−0.04–0.52)			0.058
HADS depression, *n* = 731, 632, 99																							0.135 ^3^
None (≤7)	333 (45.6)		528 (72.2)			469 (64.2)		293 (46.4)		458 (72.5)			420 (66.5)		40 (40.4)		70 (70.7)			49 (49.5)			
Mild (8–10)	188 (25.7)		122 (16.7)			147 (20.1)		164 (25.9)		110 (17.4)			116 (18.4)		24 (24.2)		12 (12.1)			31 (31.3)			
Severe (11–21)	210 (28.7)		81 (11.1)			115 (15.7)		175 (27.7)		64 (10.1)			96 (15.2)		35 (35.4)		17 (17.2)			19 (19.2)			
Change (1–2) *p*-value Effect size (CI-95%)	**<0.001** 0.62 (0.42–0.63)							**<0.001** 0.62 (0.40–0.62)							**<0.001** 0.63 (0.29–0.86)								0.983
Change (2–3) *p*-value Effect size (CI-95%)			**<0.001** −0.18 (−0.29–0.09)							**<0.001** −0.15 (−0.27–0.05)							**0.001** −0.35 (−0.68–0.12)						**0.024**
Change (1–3) *p*-value Effect size (CI-95%)						**<0.001** 0.39 (0.24–0.45)							**<0.001** 0.40 (0.24–0.46)							**0.002** 0.32 (0.02–0.59)			0.055
**(b)**																							
	**All**							**Women**							**Men**								**Difference Women vs. Men**
	**Mean (SD)** **Median (IQR)**		***p*-Value ^2^**	**Effect Size**		**Effect Size (CI-95%)**		**Mean (SD)** **Median (IQR)**		***p*-Value ^2^**	**Effect Size**		**Effect Size (CI-95%)**		**Mean (SD)** **Median (IQR)**		***p*-Value ^2^**	**Effect Size**		**Effect Size (CI-95%)**			***p*-Value ^3^**
FRI, *n* = 697, 602, 95																							
Baseline (1)	59.0 [24]							58.0 [24]							63.0 [18]								**0.020**
Change (1–2)	5.0 [15]		**<0.001**	−0.60		−0.86–−0.64		5.0 [16]		**<0.001**	−0.59		−0.84–−0.61		8.0 [15]		**<0.001**	−0.72		−1.25–−0.67			0.649
Change (2–3)	1.0 [15]		**0.002**	−0.25		−0.41–−0.20		2.0 [15]		**0.002**	−0.13		−0.28–−0.05		0.0 [12]		0.723	−0.05		−0.36–0.21			0.312
Change (1–3)	7.0 [20]		**<0.001**	−0.88		−1.08–−0.86		7.0 [19]		**<0.001**	−0.68		−0.79–−0.56		7.0 [20]		**<0.001**	−0.60		−0.86–−0.28			0.651
Godin 7 days, *n* = 706, 614, 92																							
Baseline (1)	2.0 (1)							2.0 (1)							2.0 (1)								0.747
Change (1–3)	0.0 [1]		**<0.001**	−0.26		−0.34–−0.13		0.0 [1]		**<0.001**	−0.26		−0.35–−0.13		0.0 [1]		0.008	−0.24		−0.50–0.08			0.250
Godin activity level, *n* = 468, 407, 61																							
Baseline (1)	21.0 [27]							20.0 [26]							25.0 [27]								**0.047**
Change (1–3)	2.0 [20]		**<0.001**	0.15		0.03–0.28		2.0 [19]		**<0.001**	0.17		0.03–0.31		0.0 [19]		0.877	−0.04		−0.40–0.31			0.109
EQ-5D index, *n* = 715, 620, 95																							
Baseline (1)	0.36 [0.60]							0.47 [0.60]							0.23 [0.60]								0.263
Change (1–2)	0.04 [0.34]		**<0.001**	0.43		0.32–0.53		0.04 [0.35]		**<0.001**	0.40		0.28–0.50		0.06 [0.33]		**<0.001**	0.43		0.14–0.71			0.622
Change (2–3)	0.0 [0.16]		0.132	0.00		−0.10–0.10		0.00 [0.17]		**0.041**	0.03		−0.08–0.15		0.00 [0.18]		0.317	−0.11		−0.40–0.17			0.579
Change (1–3)	−0.07 [0.40]		**<0.001**	0.44		0.27–0.48		−0.07 [0.40]		**<0.001**	0.41		0.27–0.50		−0.06 [0.50]		**<0.001**	0.30		−0.04–0.53			0.062
EQ-VAS, *n* = 714, 616, 98																							
Baseline (1)	45.0 [30]							45.0 [28]							45.0 [25]								0.174
Change (1–2)	10 [25]		**<0.001**	0.66		0.50–0.72		10 [25]		**<0.001**	0.66		0.49–0.72		10 [20]		**<0.001**	0.69		0.38–0.96			0.473
Change (2–3)	0 [20]		0.519	0.03		−0.1–0.14		0 [20]		0.266	0.05		−0.06–0.16		0 [22]		0.245	−0.11		−0.40–0.16			0.132
Change (1–3)	10 [29]		**<0.001**	0.62		0.38–0.59		10 [30]		**<0.001**	0.64		0.39–0.62		10 [26]		**<0.001**	0.50		0.11–0.68			**0.035**

(a) ^1^ Positive/negative change values indicate an improvement/deterioration. ^2^ Effect sizes: Cohen’s *d* repeated measures. ^3^ Difference between women and men at baseline, linear by linear association test. Abbreviations: IMMR, interprofessional multimodal rehabilitation; NRS, numeric rating scale; PCS, pain catastrophizing scale; HADS, hospital anxiety and depression scale. *p*-values < 0.05 presented in bold. (b) ^1^ Positive/negative change values indicate an improvement/deterioration. ^2^ Wilcoxon signed-rank test. ^3^ Mann–Whitney U test. Abbreviations: MMR, multimodal rehabilitation; SD, standard deviation; IQR, interquartile range. Effect sizes: Cohen’s *d* repeated measures. FRI, functional rating index; Godin, Godin–Shephard leisure-time physical activity questionnaire; EQ-5D, European quality of life instrument 5 dimensions; EQ-VAS, European quality of life instrument visual analogue scale. *p*-values < 0.05 presented in bold.

**Table 3 ijerph-20-05051-t003:** Change in sick leave between baseline and 1-year follow-up, for all and for women and men separately, in patients that completed the 1-year follow-up.

	All			Women			Men		
	Baseline	1-Year Follow-Up	*p*-Value	Baseline	1-Year Follow-Up	*p*-Value	Baseline	1-Year Follow-Up	*p*-Value
Sick leave, *n* (%)			**<0.001 ^1^**			**<0.001 ^1^**			**<0.001 ^1^**
Full time, 100%	149 (20.0)	96 (12.9)		122 (18.9)	77 (11.9)		27 (27.0)	19 (19.0)	
Part time, 25–75%	183 (24.5)	92 (12.3)		168 (26.0)	81 (12.5)		15 (15.0)	11 (11.0)	
No sick leave	266 (35.7)	448 (60.1)		277 (35.1)	389 (60.2)		39 (39.0)	59 (59.0)	
Missing	148 (19.8)	110 (14.7)		129 (20.0)	99 (15.3)		19 (19.0)	11 (11.0)	

^1^ Chi-square test of trend. *p*-values < 0.05 presented in bold.

## Data Availability

The data that support the findings of this study can be made available on request from the corresponding author. The data are not publicly available due to ethical restrictions.

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
