# Peer review of "Interdisciplinary Multimodal Pain Rehabilitation in Patients with Chronic Musculoskeletal Pain in Primary Care—A Cohort Study from the Swedish Quality Registry for Pain Rehabilitation (SQRP)"

_ijerph, 2023, doi:10.3390/ijerph20065051_

Round 1
Reviewer 1 Report
Please see the attachment.

Reviewer 2 Report
I would like to thank the authors for their work on a worthy topic. Overall I find the manuscript well thought out from a methodology standpoint as well as written in an easily digestible format for the readership.
One area I would like to see more information about is the differences in sites and the those that made up the teams. You mention that teams could be made up of different individuals depending on the location however, the details regarding this are lacking. I wonder if you don't see differences in some of the variables you report based on team members, location, training of team members etc. This seems like a valuable part of the treatment and may be considered a confounding variable of sorts.
Perhaps mentioning this in both the methods as well as the limitations or discussion would be helpful if you are unable to break it down from a data standpoint.
I also appreciate the conclusion section regarding the strengthening of how follow ups are handled.
